There are amendments to this paper

# Directed self-assembly of herbal small molecules into sustained release hydrogels for treating neural inflammation

Jun Zheng[1], Rong Fan[2], Huiqiong Wu[1,3], Honghui Yao[4], Yujie Yan[4], Jiamiao Liu[4], Lu Ran[5], Zhifang Sun[1], Lunzhao Yi[5], Li Dang[6], Pingping Gan[7], Piao Zheng[8], Tilong Yang[9], Yi Zhang [1,2], Tao Tang[2] & Yang Wang [2]

Self-assembling natural drug hydrogels formed without structural modification and able to act as carriers are of interest for biomedical applications. A lack of knowledge about natural drug gels limits there current application. Here, we report on rhein, a herbal natural product, which is directly self-assembled into hydrogels through noncovalent interactions. This hydrogel shows excellent stability, sustained release and reversible stimuli-responses. The hydrogel consists of a three-dimensional nanofiber network that prevents premature degradation. Moreover, it easily enters cells and binds to toll-like receptor 4. This enables rhein hydrogels to significantly dephosphorylate IκBα, inhibiting the nuclear translocation of p65 at the NFκB signalling pathway in lipopolysaccharide-induced BV2 microglia. Subsequently, rhein hydrogels alleviate neuroinflammation with a long-lasting effect and little cytotoxicity compared to the equivalent free-drug in vitro. This study highlights a direct self-assembly hydrogel from natural small molecule as a promising neuroinflammatory therapy.

---

[1] Key Laboratory of Hunan Province for Water Environment and Agriculture Product Safety, College of Chemistry and Chemical Engineering, Central South University, 410083 Changsha, China. [2] Institute of Integrative Medicine, Key Laboratory of Hunan Province for Liver Manifestation of Traditional Chinese Medicine, Xiangya Hospital, Central South University, 410008 Changsha, China. [3] Key Laboratry of Materials Processing and Mold, Ministry of Education, Zhengzhou University, 450002 Zhengzhou, China. [4] Xiangya School of Medicine, Central South University, 410013 Changsha, China. [5] Yunnan Food Safety Research Institute, Kunming University of Science and Technology, 650500 Kunming, China. [6] Department of Chemistry and Key Laboratory for Preparation and Application of Ordered Structural Materials of Guangdong Province, Shantou University, 515063 Shantou, China. [7] Department of Oncology, Xiangya Hospital, Central South University, 410008 Changsha, China. [8] College of Integrated Traditional Chinese and Western Medicine, Hunan University of Chinese Medicine, 410208 Changsha, China. [9] Southern University of Science and Technology, 518055 Shenzhen, China. These authors contributed equally: Jun Zheng, Rong Fan. Correspondence and requests for materials should be addressed to Y.W. (email: wangyang_xy87@csu.edu.cn) or to Y.Z. (email: yzhangcsu@csu.edu.cn) or to T.T. (email: tangtaotay@csu.edu.cn)

Natural small molecules are regarded as promising drug resources due to their wide range of pharmacophores and high degrees of stereochemistry[1,2]. Unfortunately, clinical treatments using these well-studied natural products are limited owing to the poor solubility and unsatisfying stability[3,4]. For years, scientists focus on hydrogel-based drug delivery systems to improve the solubility and stability of natural small molecules[5,6]. Numerous supramolecular hydrogels containing drug complexes have successfully been developed in the laboratory for inflammatory alleviation[7,8], wound repair[9], bacteria resistance, and tumor inhibition[10,11]. However, clinical researchers continually find that the involvement of drug carriers may lead to poor biocompatibility and biodegradability, low loading efficacy, and potential side effects. Additionally, these biological materials from drug carriers require complex syntheses and are relatively expensive[12], which create great obstacles for effective clinical application. Thus, chemists and pharmacologists persistently desire to prepare directed self-assemble hydrogels, referring to self-delivering, self-releasing, stable, injectable and stimuli-responsive hydrogels without any structural modification and delivery cargoes. The hydrogels are expected to be capable of superior solubility, optimal therapeutic efficacy, and almost no cytotoxicity.

Abundant innovative efforts on the direct self-assembly biological hydrogels of small molecules (especially lanreotide, diphenylalanine, Fmoc-diphenylalanine, and curcumin) have been explored[13–15]. Many self-assembly hydrogel systems derived from natural products following structural modification have been invented (such as taxol[16,17], camptothecin[18], and dexamethasone[19]). Despite these endeavors, designing directed self-assemble hydrogels formed by natural small molecules is still a formidable challenge. The exploration still largely relies on serendipitous, because the construction requires a perfectly stable equilibrium among a series of complicated and meticulous balances, including amphipathicity, intermolecular interactions, chirality, and spatial molecular arrangement[20].

We have been continuously working on rhein (the chemical structure is shown in Fig. 1a), an anthraquinone mainly isolated from the traditional Chinese medicine rhubarb (*Rheum palmatum* L. or *Rheumtanguticum* Maxim, Dahuang in Chinese) for at least a decade[21]. Rhein performs neuroprotection via anti-inflammation in treating cerebral injuries including neurodegenerative diseases and traumatic brain injury[22,23]. Nevertheless, the solubility of rhein remains poor and simultaneously exhibits low bioavailability by metabolism of glucuronidation in liver[24,25], resulting in a hindrance to clinical transformation. To enhance the therapeutic efficacy and minimize negative effects, a few efforts to prepare polymeric microparticles and nanoparticles containing rhein have been attempted[26,27]. However, drug loss during the fabrication process and premature release of payload still lead to lower drug loading and adverse systemic toxicity[3].

We believe that directed self-assembly of rhein should be a promising solution. In this study, rhein directly self-assembles into a supramolecular hydrogel via intermolecular π-π interactions and hydrogen bonds. The rhein hydrogel has excellent biostability, sustained drug release, and reversible stimuli-responsive performances. In particular, the as-prepared rhein hydrogel exerts better anti-neuroinflammation than its free-drug form with almost no cytotoxicity. To investigate such superior anti-neuroinflammatory effects, we explore the underlying molecular mechanisms. We demonstrate that the rhein supramolecular hydrogel is easier to enter cells than free-drugs, and increases accumulation to intensively bind to the active site of toll-like receptor 4 (TLR4). These properties achieve optimal anti-inflammation through inhibition of the TLR4/NFκB signaling pathway, which essentially boosts the therapeutic efficacy and reduces the negative effects. These features trigger the rhein hydrogel to serve as a promising therapeutic agent for anti-neuroinflammation.

## Results

**Morphology and gelation properties of rhein hydrogel.** In this study, the rhein direct self-assembly hydrogel was formed in PBS buffer by simple ultrasound with heating to obtain a homogeneous solution and subsequently cooled to room temperature. The minimum gelation concentration was 14.1 mM (Supplementary Fig. 1). The appearance of the hydrogel was a uniform orange-red. The scanning electron microscopy (SEM) images revealed that the scaffold was a 3D network composing of nanofibers (Fig. 1b). As can be observed in the transmission electron microscopy (TEM) and atomic force microscope (AFM) images, the nanofibers had an average diameter of approximately 30 nm with several micrometers in length (Fig. 1c and Supplementary Fig. 2).

Experimental tests indicated that pH intensively influenced the direct self-assembly process. The optimal pH values of the gel were between 8.0 and 9.4 (Supplementary Fig. 3a). Under this environment, we produced a translucent hydrogel. When the value of pH was above 9.4, the hydrogel began to collapse and became a blood red solution. The SEM images confirmed that the solution possessed a short, ribbon-like structure (Supplementary Fig. 3b). Within the ranges between pH 8.0 and pH 6.8, the sample was a viscous gel rather than a translucent gel. The microscopic morphology predominantly exhibited long fibers, while there were several short fibers distributed on the surface (Supplementary Fig. 3c). Once the pH values were set under 6.8, we observed precipitate formation. The SEM images displayed that these precipitates were the short, rod-like-structures (Supplementary Fig. 3d). These results suggest that the formation of the hydrogel is highly dependent on the degree of carboxyl deprotonation of rhein molecules. To evaluate the role of the carboxyl group in gel formation, we sought to prepare the other anthraquinones from rhubarb, including emodin, chrysophanol, aloe-emodin, and physcion (Supplementary Fig. 4). We found that none of these anthraquinones formed hydrogels under the conditions tested, demonstrating that the carboxyl group plays an important role in the gelation process.

The rhein hydrogel showed multi-responsiveness to external environment changes (Fig. 1d). When the temperature increased to 40 °C, the hydrogel turned into solution. Once the temperature was adjusted to room temperature, the sol-to-gel phase transition occurred after several hours. In addition, the gel–sol transitions were triggered through vigorous shaking by hand[28,29], and the gel state was restored after standing within several minutes. Furthermore, the reversible properties of the rhein hydrogel were investigated by oscillatory shear rheology. When the strain exceeded 23.74%, the loss modulus (G″) was higher than the storage modulus (G′), revealing the transition from the gel to the solution state (solution state: G′ < G″, gel state: G′ > G″) (Fig. 1e). The step-strain test (Fig. 1f) revealed that G′ was higher than G″ at low strain 0.1%, and G′ was lower than G″ at higher strain 35%. The material properties of the rhein hydrogel recovered when transitioning from high strain to low strain, displaying the injectable and self-healing properties of the rhein hydrogel[30,31]. We also performed rotated rheology measurement to examine the viscoelastic properties. The dynamic frequency sweep showed that the values of G′ were 5 times larger than those of G″ (Fig. 1g). Moreover, there was a weak dependency with frequency. Additionally, the G′ was much greater than the G″ during the entire process, according to the dynamic time sweep data (Fig. 1h)[32]. The excellent thixotropy and thermo-reversibility indicate that the rhein hydrogel can serve as an injectable hydrogel[33].

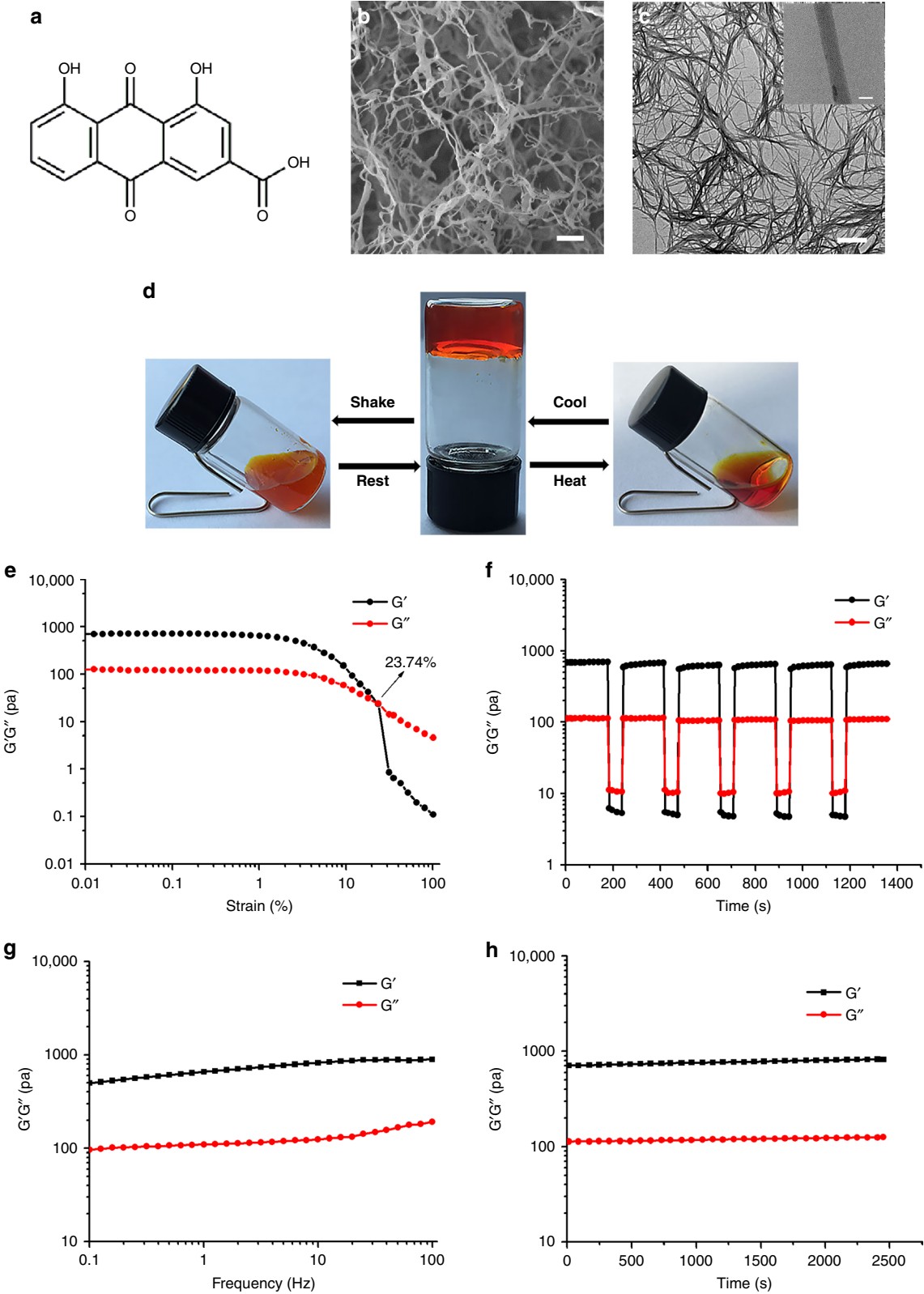

**Fig. 1** Molecular formula of rhein and characterization of the rhein hydrogel. **a** Chemical structures of rhein and digital images of the rhein hydrogel. **b** SEM image of the rhein hydrogel, scale bar: 2 μm. **c** TEM image of the rhein hydrogel, scale bar: 500 nm. The upper right is a partial enlarged view, scale bar: 20 nm. **d** Reversible gel–sol transitions of rhein hydrogel triggered by shear stress and temperature. **e** Strain-dependent oscillatory shear rheology of the rhein hydrogel with a fixed frequency of 10 rad s$^{-1}$. **f** Step-strain measurements of the hydrogel over five cycles at low strain (0.1%) and high strain (35%), frequency 10 rad s$^{-1}$. **g** Dynamic frequency sweep of the rhein hydrogel, measured at 0.1% strain. **h** Dynamic time sweep of the rhein hydrogel, at the strain of 0.1% and at the frequency of 10 rad s$^{-1}$. The concentration of hydrogel was 5 mg mL$^{-1}$ (17.6 mM), T = 25 °C

**The directly self-assemble mechanism of rhein hydrogel**. we used the mass spectrometry (MS) system to explore the structure of rhein aggregates, which is a powerful tool to investigate the assembly of aggregates[34]. The monomers ($m/z$ 283.0249), dimers ($m/z$ 589.0388), trimers ($m/z$ 895.0514), tetramers ($m/z$ 1201.0647) and higher-order aggregates were observed from MS (Fig. 2a, b). These aggregates existed as sodium salt clusters, the dimers were the predominant aggregate ions, indicating that the dimer is relatively stable relative to other aggregates. These results were further proved by the MS/MS analysis. We found the fragment ions of the dimer, rhein sodium salt ($m/z$ 305.0061) and rhein monomer from the MS/MS spectrum of higher-order aggregates (Fig. 2c–g). These results demonstrate that rhein monomers and sodium salt of rhein are the building block of rhein nanofibers[35].

To shed light on the self-assembly process of rhein fibrils, we implemented spectroscopy studies. The FT-IR results suggested that there was a typical hydrogen bond absorption peak of O-H at 3436 cm$^{-1}$. During the gelation, the peak of O-H at 3436 cm$^{-1}$ shifted to 3429 cm$^{-1}$, simultaneously accompanied by an enhanced intensity in the hydrogel phase. On the other hand, the peak of C=O at 1704 cm$^{-1}$ shifted to 1662 cm$^{-1}$ (Supplementary Fig. 5). These shifts indicate that hydrogen bonds are involved in the self-assembly process[36,37].

The concentration-dependent UV/Vis absorption study displayed the characteristic anthraquinone group peak, which underwent a redshift from 418 to 503 nm as the concentration of rhein increased (Fig. 3a), which was assigned to the π-π interactions of the anthraquinone structure. This phenomenon suggests that the aggregate configuration changed from H-type aggregation to J-type aggregation[38–40].

We next performed fluorescence spectroscopy analysis to assess whether π-π stacking played a role during the self-assembly process. As shown in Fig. 3b, an emission peak centered at 601 nm appeared in the solution, while the center of the emission peaks from the rhein hydrogel at 601 shifted to 616 nm. As the concentration of the hydrogel increased, the fluorescence intensity decreased and the peak red shifted (Supplementary Fig. 6). The fluorescence tended to quench due to the aggregation of rhein molecules. The peak redshift was attributed to the presence of the anthraquinone excimert which mainly contributed to the self-assemble process through π-π interactions[39,41]. X-ray diffraction (XRD) also confirmed this result. The apparent peak at $d = 3.4$ Å was a typical distance of π-π stacking interactions between the two moleculars (Fig. 3c)[42].

We further employed circular dichroism (CD) spectra to explore the molecular packing of rhein within the hydrogel. The CD spectra exhibited strong negative peaks at 228 nm and 412 nm (Fig. 3d). These observations suggest that the chiral packing of rhein within the hydrogel is formed in the self-assembly process. The negative peaks revealed the left-hand helical arrangement. With the concentration of rhein hydrogel increased, the CD spectra displayed a strong negative bisignate CD signal between 210 and 300 nm. These CD signal peaks showed an obvious redshift. The shift phenomena show that the J-aggregation of rhein occurs in the rhein hydrogel[43].

Combining experimental results with theoretical calculation, we proposed a possible self-assembly process of rhein hydrogel. The gelation of rhein was highly dependent on pH. When the pH value was set between 8.0 and 9.4, the rhein hydrogel was obtained. Under this condition, certain molecules were deprotonated to form rhein sodium salt. Rhein monomer and rhein sodium salt were together assembled to form dimer and higher-order aggregates. The two molecules were arranged in a J-type aggregation manner by π-π stacking and hydrogel bonding to form a dimer. Due to the electrostatic repulsion between the

carboxylic acid ions, two molecules tended to be arranged in opposite direction. The density functional theory (DFT) calculations also supported this conformation (Supplementary Fig. 7). Subsequently, dimers were further assembled into trimers, tetramers and higher-order aggregates. The rhein molecules continually added to the per-existing aggregates in a left helix fashion, resulting in the formation of the nanofiber with a left-handed helical configuration. The nanofibers further crosslinked to form 3D networks (Fig. 3e).

**Assessments of sustained release and cytotoxicity**. We examined the sustained release of rhein hydrogel within a series of concentrations. Figure 4a confirmed the release profile. There was a fast release during the first 12 h, and then the release of rhein showed a gradual process. Regardless of high concentration or low concentration, the release rate was up to 70% after 24 h, which was attributed the good solubility of the hydrogel and the brittleness of the fibers[7]. As the gel concentration increased, the release rate decreased, accompanied by the extension of the release time. Moreover, the cumulative release percentage at low concentrations reached 90% within 36 h. Only a small amount of rhein was released from 36 to 72 h. The cumulative release rate under high concentrations merely came to 80% within 36 h and the release rate reached 88% after sustained released to 72 h. The higher concentration of rhein hydrogel which possessed a closer fibers network, resulted in more durable release of the rhein molecules from hydrogel. These properties facilitate the rhein hydrogel to serve as a potential biomedical application.

Stability is a key indicator to evaluate whether a drug can potentially be used in clinical application. The rhein hydrogel maintained a good stability after being stored at room temperature for 3 months (Supplementary Fig. 8). In order to investigate the stability of the rhein hydrogel in cells. We used liquid chromatography-mass spectrometry (LC-MS) to detect the concentration of rhein in the BV2 cells treated with rhein hydrogel and free-drug for 3 h, 12 h, 24 h, 48 h, and 72 h, respectively. As shown in Fig. 4b, the concentration of the rhein from hydrogel group was higher than the free-drug group at each time point. Furthermore, the dimers were merely observed in rhein hydrogel group (Supplementary Fig. 9). These data suggest that rhein nanofiber is more likely to penetrate into cells than free-drug form. In addition, we found that the concentration of free-drug reached the top at 24 h and then decreased at 48 h. While there was a plateau during 24–48 h in rhein hydrogel group, indicating that the rhein hydrogel is more stable in the cells.

To evaluate the toxicity, we executed the 3-(4,5-dimethylthia-zol-2-yl)-2,5-diphenyltetrazolium bromide (MTT) assay by assessing the effect of the drug concentration on cell viability. Figure 4c suggested that the rhein hydrogel and the equivalent free-drug dosing showed no obvious cytotoxic effects on BV2 cells within the ranges of 5-40 μM at 24 h. When the treatment continued to 48 h (Fig. 4d), 40 μM of the free-drug significantly diminished the BV2 cells viability by 22% compared with the rhein hydrogel. Interestingly, the rhein hydrogel at various concentrations did not affect the viability of BV2 cells. The results demonstrate that the cytotoxicity of the rhein hydrogel is lower than that of the free-drug during the slow and sustained release[44].

**Rhein hydrogel exerts better anti-neuroinflammation**. For medical transformation, we used a LPS-induced BV2 cell line as an in vitro model to estimate the anti-neuroinflammatory effects of the rhein hydrogel. During neuroinflammation, brain microglia underwent morphological changes and proliferation, triggering the excess production of various proinflammatory

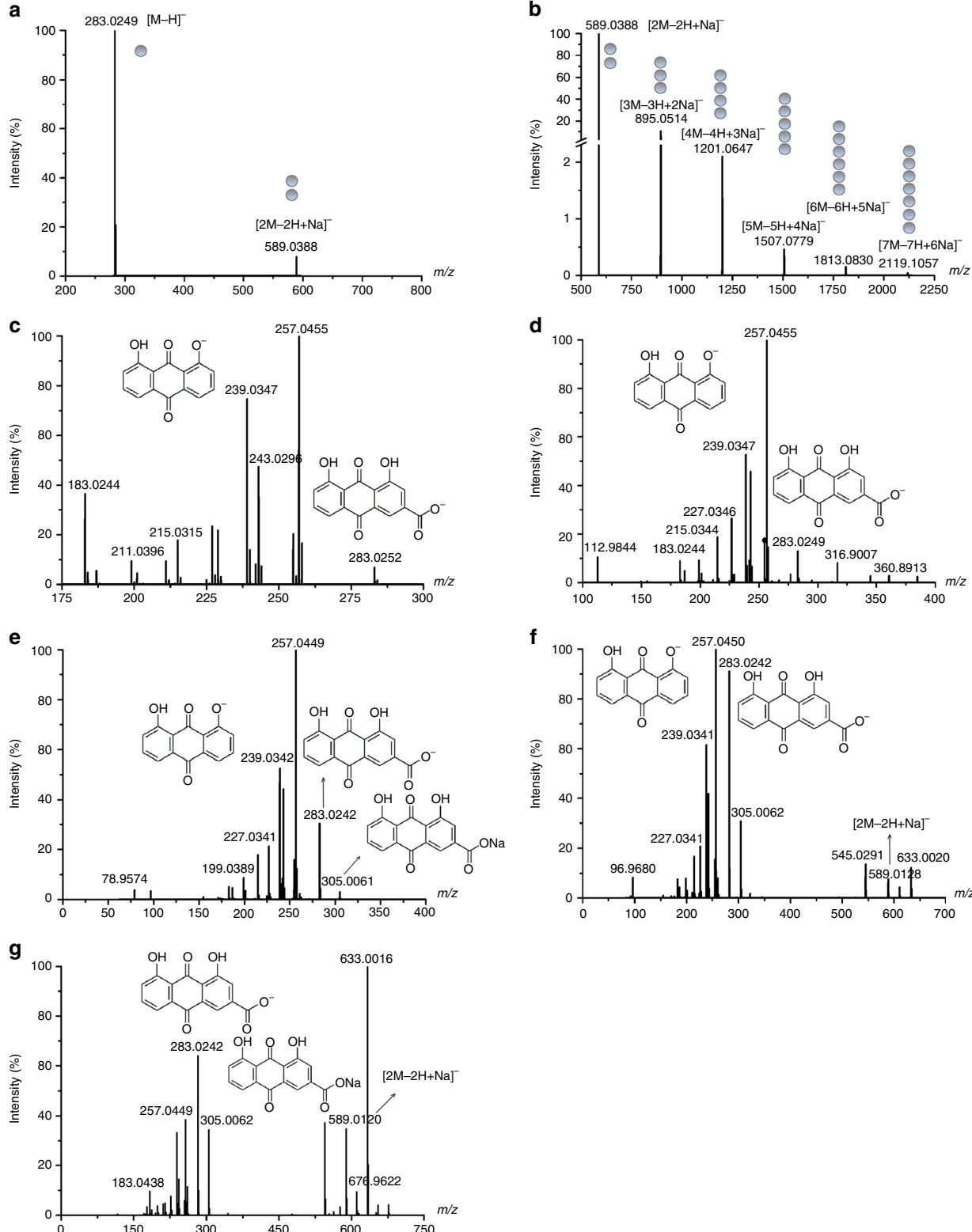

**Fig. 2** Mass spectrometric analysis of the rhein hydrogel. **a**, **b** MS analyses of the rhein hydrogel. **c** MS/MS analyses of monomer [M−H]⁻. **d** MS/MS analyses of dimer [2M−2H+Na]⁻. **e** MS/MS analyses of trimer [3M−3H+2Na]⁻. **f** MS/MS analyses of tetramer [4M−4H+3Na]⁻. **g** MS/MS analyses of pentamer [5M−5H+4Na]⁻

mediators. TNF-α and IL-1β have been recognized as the key factors in neuroinflammation. Microglia cells are primed by neuroinflammatory disorders to produce exaggerated responses to IL-1β and TNF-α[45]. Thus, we primarily focused on the drug

effects on these two cytokines. We tested the rhein hydrogel in a therapeutic setting in LPS-stimulated BV2 cells. LPS led to the marked accumulation of TNF-α and IL-1β compared to the control (Fig. 5). After 24 h treatment, the hydrogels and the

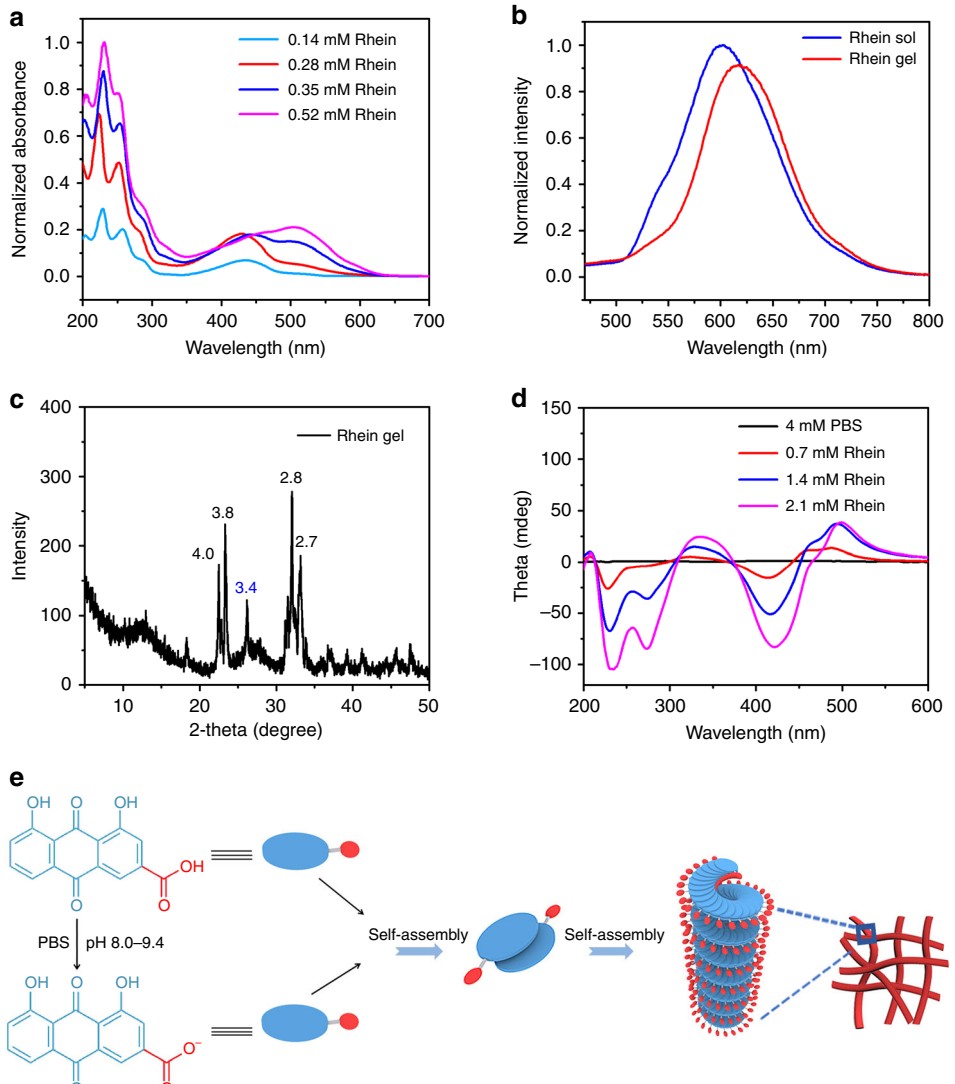

**Fig. 3** The self-assemble mechanism of the rhein hydrogel. **a** UV/Vis spectrum of the rhein hydrogel. **b** Fluorescence emission spectra of rhein solution and hydrogel. The samples were recorded at 4 mg mL$^{-1}$ (14.1 mM). **c** XRD pattern of the rhein xerogel (the unit of inserted distances is Å). **d** CD spectra of the rhein hydrogel at different concentrations. **e** Self-assembly process diagram of the rhein hydrogel

equivalent free-drug dosing resulted in the obvious elimination of TNF-α and IL-1β, as confirmed by the hypointensity of red (TNF-α) and green (IL-1β) fluorescent protein-positive cells co-expressed in the microglia kytoplasm (Fig. 5a). Similarly, subsequent western blotting and enzyme linked immunosorbent assay (ELISA) revealed that the levels of TNF-α and IL-1β were dramatically reduced with both drugs compared to the LPS-treated group (Fig. 5b–f). In addition, the results of ELISA assays showed that both drugs notably decreased the releases of IL-6, IL-12, and iNOS compared to the LPS-treated group (Fig. 5g–i).

To observe the sustained release efficacy, the treatment continued for 48 h. We found that the hydrogel significantly lowered the levels of TNF-α, IL-6, IL-12, and iNOS compared with the equivalent free-drug dosing. (Fig. 5). The observations reveal that the rhein hydrogel exhibits better neuroinflammatory prevention than the free-drug through the direct self-assembly nanofibers under sustained release control.

**Rhein hydrogel enhances TLR4/NFκB inhibition**. To further elucidate the anti-inflammatory pathways of the rhein hydrogel, it is critical to explore the underlying molecular mechanisms.

Neuroinflammation is mainly implicated in the activation of p38, PI3K/Akt and TLR4/NFκB signaling pathways[46–48]. We used western blotting analyses to investigate the anti-neuroinflammatory mechanisms of rhein hydrogel and the free drug. We studied the effects of rhein and rhein hydrogel on TLR4/NFκB signaling pathways. Following activation of BV2 cells by LPS, increasing TLR4 expression followed by the enhancement of phosphorylation and degradation of IκBα in the cytoplasmic extracts were observed compared to the control (Fig. 6b–e). NFκB activity was evaluated according to the nuclear translocation of the p65 subunit of NFκB. The findings revealed that LPS induced phosphorylation of IκBα, further resulting in the remarkable shift of NFκB p65 into the nucleus. The presence of rhein hydrogel significantly prevented increases in TLR4, phosphorylation of IκBα and nuclear translocation of p65. In particular, during the phase of sustained release (48 h), the rhein self-assembly hydrogels better inhibited the TLR4/NFκB pathway than did the equivalent free-drug dosing (Fig. 6b–e). We also surveyed the effects of the rhein hydrogels and free-drug on the other neuroinflammatory signaling pathways, including p38 and PI3K/Akt (Supplementary Figs. 10 and 11). The results indicate that no

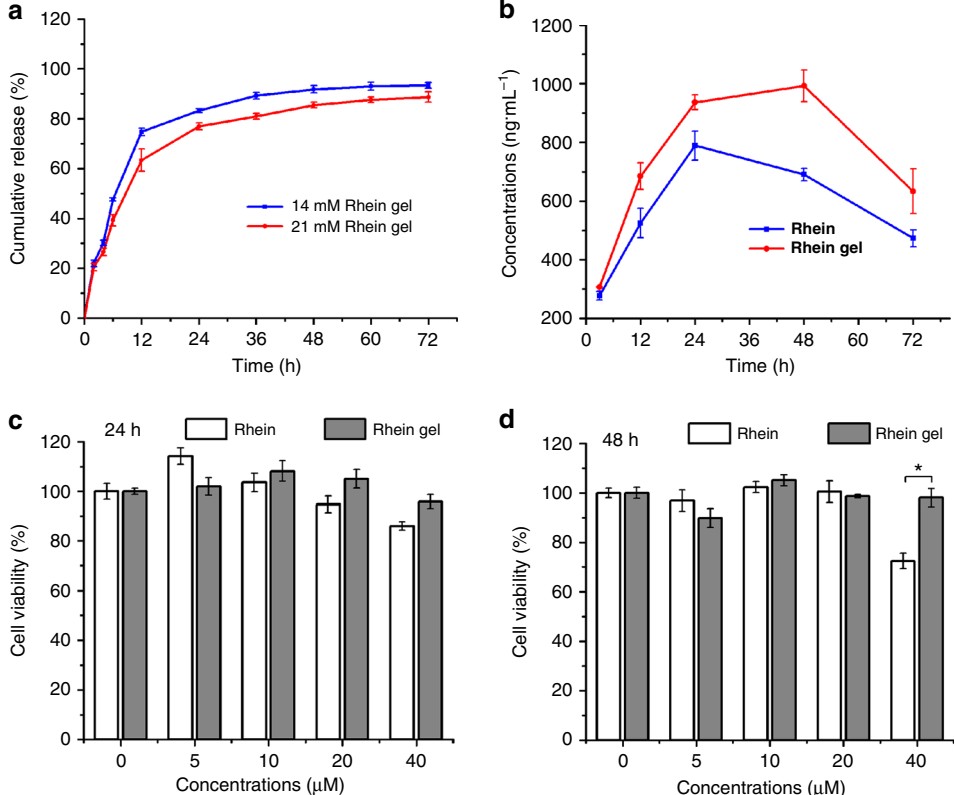

**Fig. 4** Rhein hydrogel performed better sustainable release and lower cytotoxicity. **a** In vitro release profiles of rhein hydrogel, the concentrations of rhein hydrogel were 4 mg mL$^{-1}$ (14.1 mM) and 6 mg mL$^{-1}$ (21.1 mM), respectively. The gels were incubated with PBS buffer solution (pH 7.4) at 37 °C. **b** The curve of the concentration of rhein in the cells over time. **c**, **d** Cell viability (24 h and 48 h) of BV2 cells incubated with different concentrations of rhein hydrogel and rhein, the concentrations of the samples range from 0 to 40 μM. Values are expressed as the means ± SEM ($n = 3$). One-way ANOVA with Tukey's post hoc test. *$p < 0.05$. Error bars represent SEM

better inhibitions of p38 and PI3K/Akt phosphorylations induced by the rhein hydrogel were found compared to the free drug at 48 h. Our work demonstrates that the better therapeutic effects induced by the rhein hydrogel under sustained release are mainly associated with TLR4/NFκB inhibition.

## Discussion

In this work, we have presented rhein-based direct self-assembly hydrogel. The rhein hydrogel exhibits excellent solubility, sustained release properties, low toxicity, and reversible stimuli-responsive properties to shear stress and temperature. As a directed self-assemble drug hydrogel, the rhein hydrogels significantly enhance anti-neuroinflammation compared with the free-drug through downregulation of proinflammatory cytokines and neurotoxic factors. The superior sustained release anti-neuroinflammatory effects from the rhein self-assembly hydrogels are associated with the inhibition of the TLR4/NFκB signaling pathway. The directed self-assemble hydrogels display superior solubility, optimal therapeutic efficacy, and fewer adverse effects. These properties facilitate the rhein hydrogel to serve as a potential anti-neuroinflammatory agent.

The gelation pathway is of key importance to append this unique gel system to other natural small molecules with low solubility. Our experiments indicate that the rhein hydrogel was formed by the self-assembly of rhein molecules through π-π stacking, hydrogen bonds, and other non-covalent interactions. Based on the experimental results, a possible self-assembly mechanism of the rhein hydrogel was proposed. The evidence demonstrates that the rhein monomer is capable of

accomplishing self-assembly processes without any structural modification or delivery carrier. This provides a potential guide for the self-assembly of other natural small molecules.

Understanding the anti-neuroinflammatory effects of the rhein hydrogel is the core of this study. It is notable that with treatment of free-drug and hydrogels for 24 h, both drugs exhibit similar anti-neuroinflammatory effects without obvious cytotoxicity by reducing proinflammatory mediators, including TNF-α, IL-6, IL-12, and iNOS. More interestingly, as the treatment continues for 48 h, the rhein diminishes microglial cell viability at 40 μM, whereas the rhein hydrogel does not affect cell viability. Furthermore, the rhein hydrogel markedly lowers the levels of TNF-α, IL-6, IL-12, and iNOS compared with the rhein monomer. The results reveal that the rhein hydrogel provides less cytotoxicity and better neuroinflammatory prevention than the free-drug under the sustained release control.

It is critical to explore how the rhein hydrogel interacts with protein targets to achieve the better therapeutic effects. Our work illustrates that both the rhein hydrogel and the free-drug block p38, PI3K/Akt, and TLR4/NFκB pathways to alleviate neuroinflammatory responses. Particularly, the better therapeutic effects induced by the rhein hydrogel under the sustained release are primarily associated with TLR4/NFκB inhibition, rather than p38 and PI3K/Akt pathways. This distinction may be attributed to the nanofiber structure of the rhein hydrogel. Moreover, we further use molecular docking techniques to elucidate the underlying mechanism. We confirm that TLR4 is a potential druggable target of rhein (Fig. 6a). This specific recognition facilitates the transfer of LPS to MD-2 protein, further activating the downstream components, including NFκB, and leading to the excess releases

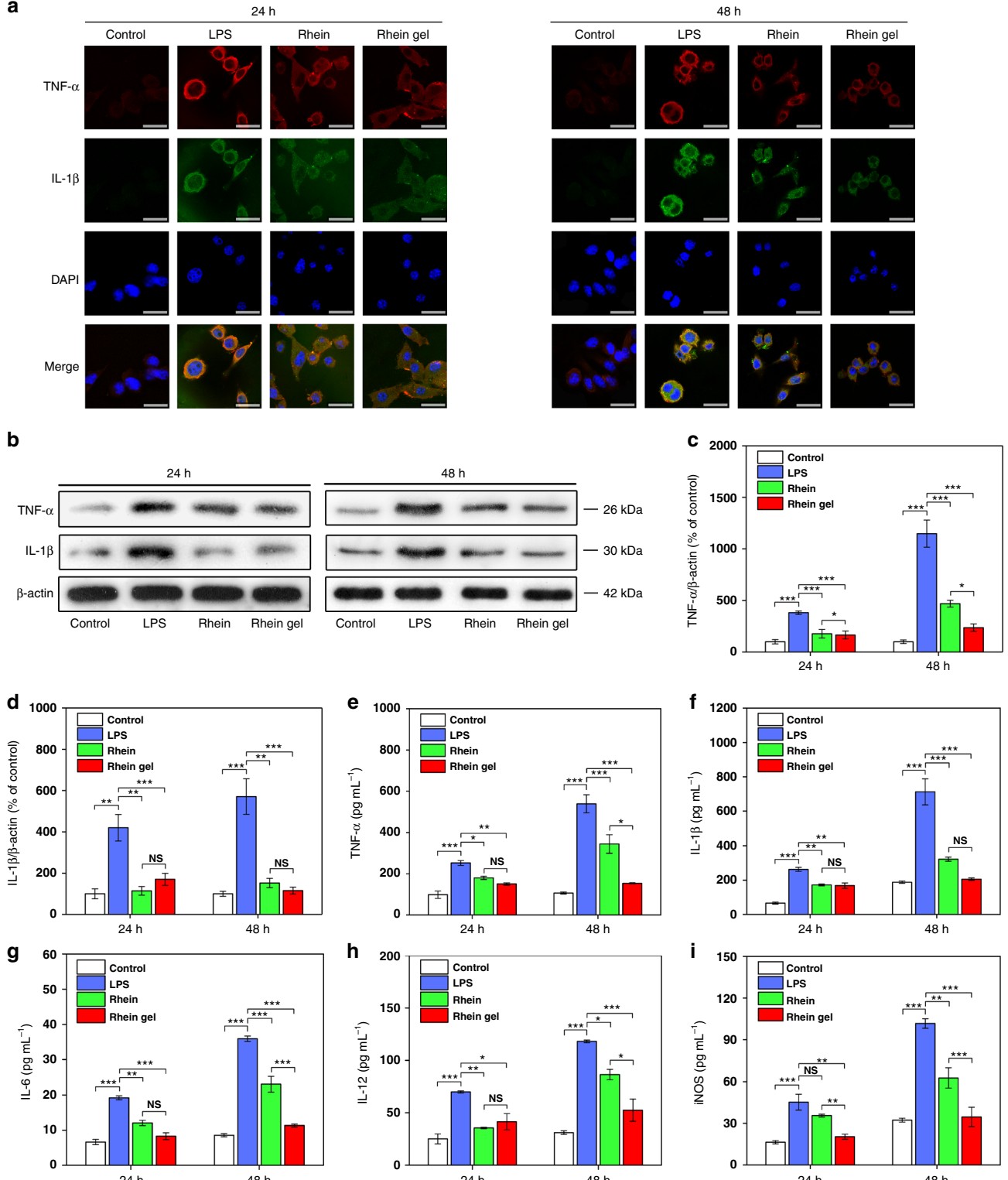

**Fig. 5** Rhein and rhein hydrogel alleviated neuroinflammation in LPS-induced BV2 cells. **a** Confocal microscopy images from BV2 cells (treated with control, LPS, rhein and rhein hydrogel). Immunofluorescence staining is shown. Cell nuclei are shown in blue (DAPI), TNF-α is shown in red, IL-1β is shown in green, and yellow labeling represents co-localization. Scale bar, 15 μm. **b** Representative western blots of TNF-α and IL-1β proteins at 24 h and 48 h. The uncropped and unprocessed scans of blots are presented in Supplementary Fig. 12. **c** Quantifications of TNF-α at 24 h and 48 h. **d** Quantifications of IL-1β proteins at 24 h and 48 h. **e–i** ELISA assays for determinations of TNF-α (**e**), IL-1β (**f**), IL-6 (**g**), IL-12 (**h**) and iNOS (**i**) levels at 24 h and 48 h. Values are expressed as means ± SEM ($n = 3$). One-way ANOVA with Tukey's post hoc test. *$p < 0.05$, **$p < 0.01$, ***$p < 0.001$. Error bars represent SEM

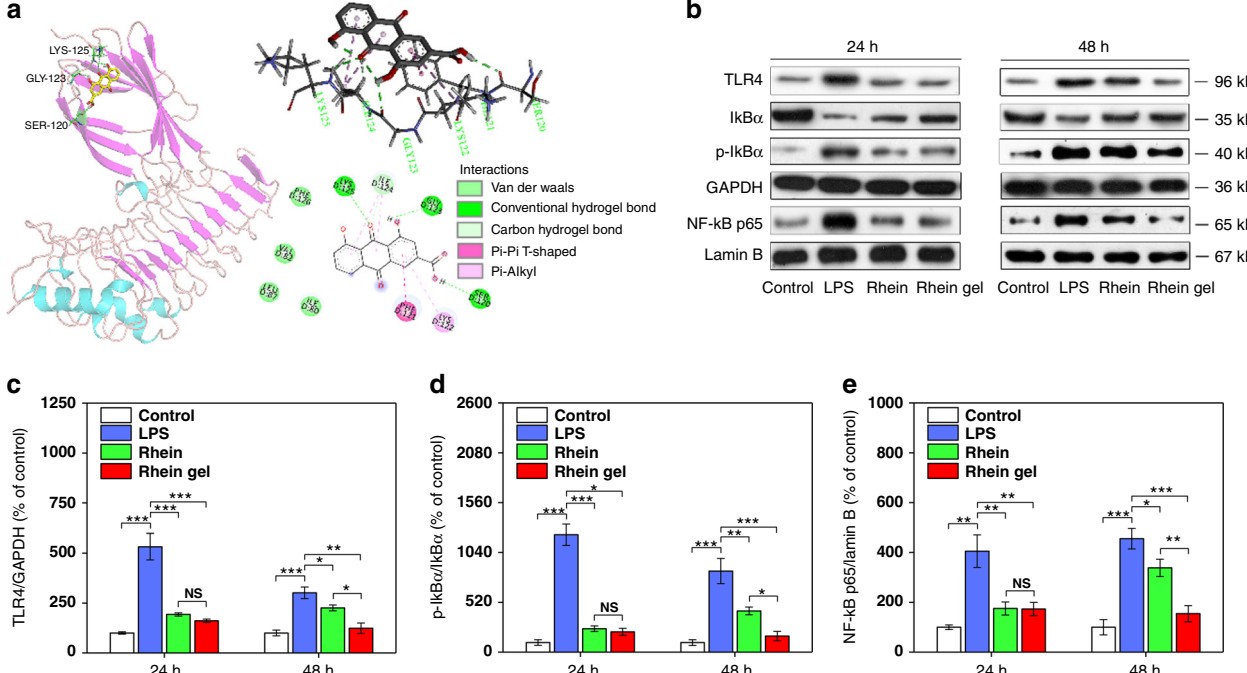

**Fig. 6** Rhein and rhein hydrogel inhibited TLR4/NFκB pathway in LPS-induced BV2 cells. **a** Molecular docking analysis of the interaction between rhein and TLR4. Molecular docking model illustrated the interaction between rhein and the active sites of TLR4, showing the existences of intermolecular hydrogen bonds and π-π interactions. **b** Rhein and its hydrogel blocked the TLR4/NFκB pathway, as shown by western blot analysis. Representative western blots of TLR4 in total proteins, IκBα and p-IκBα in cytosolic extracts, and p65 in nuclear extracts after treatment with rhein or rhein hydrogel for 24 and 48 h. The uncropped and unprocessed scans of blots are presented in Supplementary Fig. 13. **c–e** Quantification of TLR4 (**c**), p-IκBα/IκBα (**d**) and p65 (**e**) according to densitometric analysis. The results are presented as a ratio of TLR4/GAPDH, p-IκBα/IκBα and p65/Lamin B. Values are expressed as the means ± SEM ($n = 3$). One-way ANOVA with Tukey's post hoc test. *$p < 0.05$, **$p < 0.01$, ***$p < 0.001$. Error bars represent SEM

of proinflammatory cytokines and neurotoxic fators[49–51]. Therefore, the TLR4/MD-2/LPS complex represents a symmetrical manner to stimulate the proinflammatory responses. Inhibiting TLR4 dimerization is considered a key maneuver to regulate neuroinflammatory disorders[52,53]. This work reveals that strong hydrogen bonding and π-π interactions exist between rhein and the residues of TLR4, including Ser120, Phe121, Gly123, Ile124, Lys125, and Lys122 (Fig. 6a). The rhein binds to MD-2 (subunit of TLR4 receptor) and forms a steric hindrance effect, which results in the inability of TLR4 to efficiently identify LPS, thereby inhibiting the NFκB signaling pathway. Compared to the free drug, rhein hydrogel consists of nanofibers with relatively larger size and orderly structure, which has a strong tendency to prolong the circulation time and allow for increased accumulation[54,55]. On the other hand, the nanofibers are more easily uptaken by cells than the free-drug form[56,57], providing more opportunity for these molecules to bind to TLR4. Furthermore, more hydrogen bonds and π-π interactions from the aggregates contribute to the tight binding of TLR4. Hence, this hydrogel remarkably obscures the active site of TLR4 and blocks the access of the substrate, leading to better long-lasting effects and lower toxicity[58]. This illustration is in accord with the results of western blot in this study. Once LPS binds to TLR4 on the surface of microglia, it causes the translocation of p65 into the nucleus and binds to the DNA binding site. This regulates the transcription of its target genes, triggering the expression of pro-inflammatory enzymes and cytokines. The inhibition of TLR4 by the rhein nanofiber deactivates the downstream NFκB signaling pathway, leading to the downregulation of inflammatory mediators and cytokines. The present study shows that the superior inhibition of TLR4/NFκB at 48 h may result from the slow dissolution and continuous release of the rhein molecules

from the nanofibers. Accordingly, the rhein hydrogel can be used as a neuroinflammation-targeting agent.

In summary, we have explored direct self-assembly hydrogels of a natural product without any delivery carriers or structural modifications through hydrogen bonds, π-π interactions and electrostatic interactions. The rhein hydrogel exhibits excellent stability, sustained release properties and reversible stimuli-responsive properties. The hydrogel consists of a 3D nanofiber network structure that prevents premature degradation. Moreover, it easily enters cells, leading to an intensive binding to the active site of toll-like receptor 4. These advanced properties enable the rhein hydrogel to significantly dephosphorylate IκBα, inhibiting the nuclear translocation from the p65 at NFκB signaling pathway in LPS-induced BV2 microglia. In particular, the rhein hydrogel markedly alleviates neuroinflammation with a long-lasting effect without cytotoxicity compared with the equivalent free-drug dosing in vitro. To the best of our knowledge, this work provides a paradigm for discovering direct self-assembly hydrogels formed by natural small molecules to significantly promote therapeutic effects with no obvious cytotoxicity (Fig. 7).

## Methods
**Materials**. Rhein was obtained by Xi'an Natural Field Bio-Technique Co., Ltd. Sodium bicarbonate NaHCO₃ was provided from Sinopharm Chemical Reagent Co., Ltd. Phosphate buffer solutions (PBS) were purchased from Xiya Reagent (Chengdu, China). MTT reagent, dimethylsulfoxide (DMSO), destination access point identifier (DAPI), polyacrylamide and sodium dodecyl sulfonate were purchased from Sigma. Ultrapure water (>18 MΩ) was used. All the reagents were used without further purification throughout this study.

**Preparation of the rhein hydrogel**. Rhein was dissolved in NaHCO₃(pH 8.3, 0.2 M) or PBS (pH 8.0–9.4, 0.1 M) followed by ultrasound or heat to obtain a homogeneous solution. Subsequently, the solution was cooled to room temperature

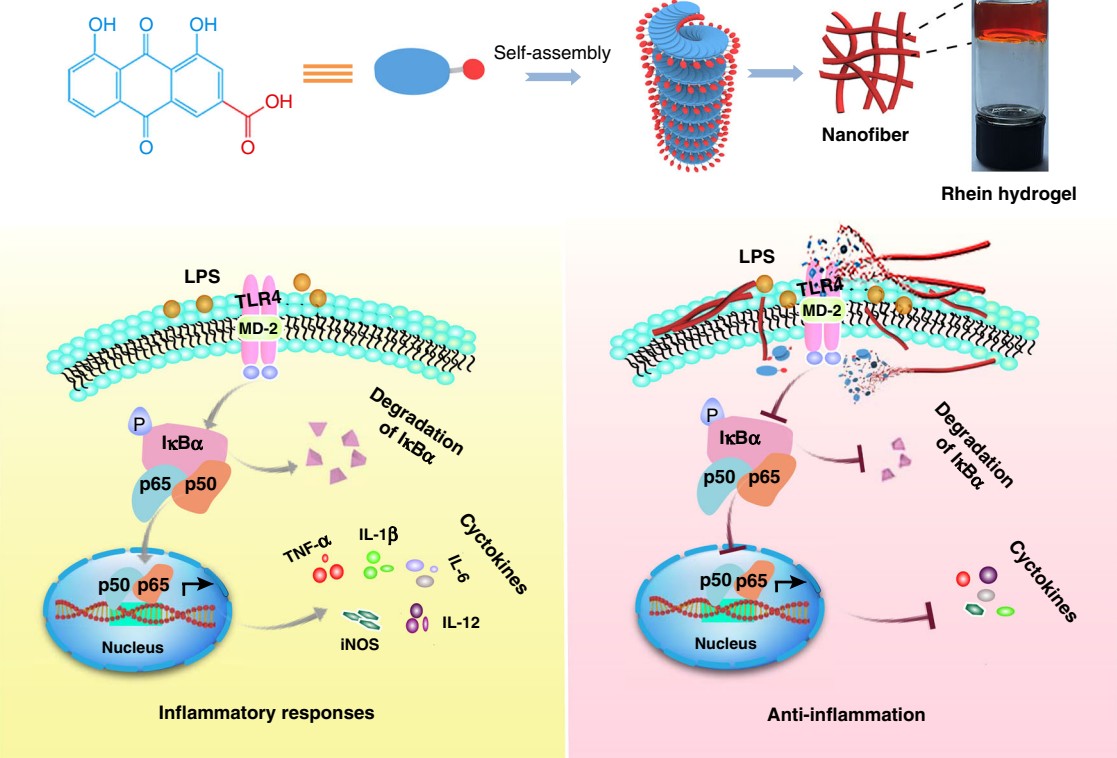

**Fig. 7** Schematic depiction of neuroinflammatory prevention induced by rhein hydrogel. Rhein is self-assembled by non-covalent interaction to form nanofibers with left-handed, which are further crosslinked to form 3D network structure. When treating inflammatory BV2 cells with rhien nanofibers, the nanofibers slowly depolymerize to release rhein or rhein aggregates. They intensively bind to MD-2 (the subunit of TLR4 receptor), obscure the active site of TLR4 and further block the access of the substrate, leading to the inhibition of NFκB activation. Subsequently, the rhein hydrogel suppresses the release of neuroinflammatory factors and mediators

and accompanied by ultrasound. A uniform stable hydrogel was obtained after 5 min, and the critical gel concentration was $4 \, \text{mg} \, \text{mL}^{-1}$ (14.1 mM).

**Rheology test**. The rheological studies were carried out on a rotated rheometer (AR 2000ex, TA Instrument, USA). The dynamic frequency sweep was measured at 0.1% strain, and the frequency was between 0.01 and 100 Hz. The dynamic time sweep was measured at 0.1% strain, and the time was kept at 2400 s. The strain-dependent oscillation was measured at 0.1–100% strain. The step-strain rheological experiment was performed in two steps. First, the rhein hydrogel was tested with a strain of 0.1% (step 1). Then, the strain was changed from 0.1 to 35% and remained at 35% for 1 min to completely damage the gels (step 2). Further, the strain was reduced from 35 to 0.1% again and remained at 0.1% for 3 min to observe whether the gel was restored. During the entire test, the frequency remained at $10 \, \text{rad} \, \text{s}^{-1}$ and the temperature was kept at 25 °C. All measurements were performed at 25 °C and at a concentration of $5 \, \text{mg} \, \text{mL}^{-1}$ (17.6 mM) hydrogel.

**Analysis of rhein aggregates by mass spectrometry**. We studied the structure of rhein nanofibers by Ultrahigh-performance liquid chromatography and mass spectrometry (UHPLC-Q/Orbitrap/MS/MS). The UHPLC-Q-Orbitrap-MS system consisted of a Dionex U3000 UHPLC system (Thermo Fisher Scientific, MA, USA) and a high-resolution Q-exactive focus mass spectrometer (Thermo Fisher Scientific, Bremen, Germany). The Dionex U3000 UHPLC system was equipped with a quaternary Series RS pump, a degasser, an auto sampler, and a column compartment. The chromatographic system was coupled to the mass spectrometer with a heated electrospray ionization source (HESI). Xcalibur 3.0 (Thermo Fisher Scientific, Bremen, Germany) was used for UHPLC-MS control and data handling.

Chromatographic separation was performed on flowing injection at 35 °C. To obtain the most remarkable separation efficiency within a short time, the mobile phase, elution conditions, and other chromatographic parameters were optimized. Consequently, gradient elution was adopted using $H_2O$ (A) and methanol (B) as a mobile phase. The isocratic elution was programmed as follows: 0–5 min, 10–100% B; The flow rate was $0.2 \, \text{mL} \, \text{min}^{-1}$, and the injection was $1.0 \, \mu L$.

*MS analysis*: The HESI parameters were optimized for accurate mass measurement as follows: sheath gas flow rate, $30 \, \text{L} \, \text{min}^{-1}$; auxiliary gas flow rate, $8 \, \text{L} \, \text{min}^{-1}$; spray voltage, 3.5 kV; capillary temperature, 320 °C; Slens RF level, 50 V; auxiliary gas heater temperature, 120 °C; and collision energy, 35 eV. Nitrogen gas was used for spray stabilization, high energy collision dissociation, and damping

gas in the C-trap. High-accuracy spectra were obtained in negative-ion mode. The mass range in the full scanning experiments was 200–3000 *m/z*.

**FT-IR spectroscopy analysis**. FT-IR spectra were recorded with a Perkin Elmer Spectrum One instrument (USA) and scanned between 4000 and $400 \, \text{cm}^{-1}$. KBr was mixed with the powdered samples to prepare the thin films. Materials were prepared at a concentration of 17.6 mM. KBr thin film was used as blank control.

**UV/Vis spectroscopy analysis**. UV/Vis spectra were recorded on a Shimadzu UV-2450 spectrometer (Japan). We collected information from 800 to 200 nm. Then 1 mL of the sample was detected by using a 10 mm aquartz cell.

**Fluorescence spectroscopy analysis**. The samples were detected using Hitachi F-7000 (Japan) in a 1 mm quartz cell. We prepared rhein solution and hydrogel with a concentration of 14.1 mM. Samples were excited at 450 nm, monitoring the emission from 470 to 800 nm.

**X-ray powder diffraction test**. XRD patterns were carried out at D/max 2550 power diffractometer (Japan) using a graphite-filtered Cu Kα radiation manipulating at 40 kV. The sample was rhein xerogel.

**Circular dichroism (CD) analysis**. Circular dichroism spectra of different samples were obtained by using Jasco-815. Approximately $200 \, \mu L$ of the sample was detected by using a 1 mm quartz cell. The bandwidth was set to 2 nm with a scanning speed of $100 \, \text{nm} \, \text{min}^{-1}$. All scans were taken from 200 to 600 nm, and all spectra were an average of 5 scans.

**DFT calculation**. The Gaussian 09 software package was used to perform geometry optimization and vibration analysis. The geometry optimization and vibration analyses were performed using M06-2X hybrid functional with all-electrons basis set 6-311 G*. The vibration frequencies were evaluated at the same level of theory that of the optimization step.

**In vitro drug release studies**. First, 1 mL of hydrogel sample (14.1 mM and 21.1 mM) with 1 mL of PBS buffer solution (pH 7.4, 0.01 M) was placed into separate dialysis bags (Viskase, MD25-2500, MW:2500) and incubated in 150 mL PBS buffer solution (pH 7.4, 0.01 M) at 37 °C. Then, 10 mL of sample was removed

at each time point and replenished 10 mL of fresh PBS solution, the sample was determined by UV/Vis absorption spectroscopy.

**BV2 microglial cells culture**. BV2 microglia cell lines (Catalogue Number: 3142C0001000000337) were purchased from National Infrastructure of Cell Line Resource (Beijing, China). BV2 microglial cells exhibit the phenotypic and the functional properties. They are the classic neuroinflammatory model. The immortalized mouse BV2 microglial cell line was purchased from National Infrastructure of Cell Line Resource (Beijing, China). The cells were grown and maintained in DMEM with 10% FBS and antibiotics (1% penicillin/streptomycin) under 5% $CO_2$ at 37 °C. The prepared cells were used for the subsequent experiments.

**Determination of rhein concentration in BV2 cells by LC-MS**. We used LC-MS to determine the concentration of rhein in the cells over time to assess the intracellular stability of rhein and rhein hydrogel. BV2 Cells were incubated in 6-well plates at a density of $2 \times 10^6$ cells for 12 h. We prepared the DMEM containing 15 μM of rhein and rhein hydrogel, respectively, and 2 mL of DMEM was added to cells. The cells were then incubated for 3 h, 12 h, 24 h, 48 h, and 72 h, respectively. Subsequently, the DMEM containing drugs was removed. 500 μL of methanol was added to each well to lyse the cells and release the compounds. The sample was collected and centrifuged at $22,000 \times g$ for 20 min after treated with a vortex shaker for 5 min. The obtained sample was detected by LC-MS.

**Cell viability measurement**. Cells were seeded in 96-well plates ($1 \times 10^4$ mL$^{-1}$ cells). The cells were then treated with a series of concentrations of drugs for 12 h. Then, 10 μL of MTT reagent (5 mg mL$^{-1}$, Sigma) was added to each well and incubated for 4 h. Afterward, the supernatants were removed. Next, 150 μL of DMSO was added to dissolve the formazan crystals. The absorbance was measured with a microplate reader (Huisong Technology Development Co., Ltd, Shenzhen, China).

**Immunofluorescence staining**. After various treatments, BV2 cells were fixed in 4% paraformaldehyde and then permeabilized with 0.3% Triton X-100. The cells were blocked with 5% bovine serum albumin (BSA). Subsequently, the cells were immunostained with primary antibodies, including TNF-α (Proteintech, 60291-1-Ig, mouse, 1:50) and IL-1β (Proteintech, 16806-1-AP, rabbit, 1:50). After incubation at 4 °C overnight, the cells were further incubated with the corresponding Alexa fluor 594-conjugated goat anti-mouse IgG (Proteintech, SA00013-3, 1:500) or Alexa fluor 488-conjugated goat anti-rabbit IgG (Proteintech, SA00013-2, 1:500) for 1 h at room temperature. Finally, nuclei were counterstained with 4′,6-diamidino-2-phenylindole (DAPI, Sigma). All images were captured using a confocal microscope (TCS SP8 X & MP, Leica, Germany).

**ELISA method**. BV2 cells ($1 \times 10^5$ cells per well in a 24-well plate) were pretreated with treatments for 1 h and stimulated with LPS (100 ng mL$^{-1}$). After treatment for 24 h and 48 h, the supernatants were collected. The concentrations of TNF-α, IL-1β, IL-6, IL-12, and iNOS in the culture medium were measured by ELISA kits, following the procedures recommended by the supplier (Cusabio, Wuhan, China).

**Western blot assay**. Total cellular and nuclear proteins were extracted from BV2 microglia cells. The protein concentrations were determined by BCA protein assay kit. Proteins were electrophoresed on 10% SDS-polyacrylamide gels, and transferred onto a PVDF membrane (Millipore, Bedford, MA). The membranes were blocked in 5% skim milk with 20% Tween20 (TBST) for 1 h at room temperature. We next used the following primary antibodies and dilutions: TNF-α (Abcam, ab66579, rabbit, 1:1000), IL-1β (Proteintech, 16806-1-AP, rabbit, 1:50), TLR4 (Proteintech, 19811-1-AP, rabbit, 1:500), IKBα (Abcam, ab32518, rabbit, 1:1000), p-IKBα (Cell Signaling Technology, #9246, mouse, 1:1000), p65 (Cell Signaling Technology, #6956, mouse, 1:1000), p38 (Proteintech, 14064-1-AP, rabbit, 1:2000), p-p38 (Abcam, ab195049, rabbit, 1:1000), PI3K (Abcam, ab40755, rabbit, 1:1000), p-PI3K (Cell Signaling Technology, #4228, rabbit, 1:1000), Akt (Cell Signaling Technology, #9272, rabbit, 1:1000), p-Akt (Cell Signaling Technology, #4060, rabbit 1:2000), β-actin (Proteintech, 60008-1-Ig, mouse, 1:4000), GAPDH (Bioworld, AP0063, rabbit, 1:2000) and Lamin B (Proteintech, 66095-1-Ig, mouse, 1:5000). The blots were incubated with the primary antibodies overnight at 4 °C. After washing with TBST, the anti-mouse secondary antibodies (Proteintech, 1:3000, SA00001-1) and the anti-rabbit secondary antibodies (Proteintech, 1:6000, SA00001-2) were incubated for 1 h at room temperature. The proteins were visualized by an enhanced chemiluminescence detection system (ThermoFisher Scientific, MA, USA). The uncropped and unprocessed scans of blots are presented in Supplementary Figs. 12 and 13.

**Molecular docking studies**. Autodock 4.0 was used in this study to evaluate the potential molecular binding mode, the PyMol molecular viewer (http://www.pymol.org/) was employed to analyze the docked structures. The crystal structure of the TLR4-MD-2 Complex with bound endotoxin antagonist Eritoran (PDB code: 2Z65) was downloaded from the RCSB Protein Data Bank (www.rcsb.org). The 3D chemical structure of rhein was retrieved from PubChem compound database (NCBI, USA) and was subjected to minimized the energy by using molecular mechanics-2 (MM2) force field in Chem 3D Pro. The protein-ligand docking active site was defined by the location of the original ligand (Eritoran).

Lamarckian Genetic Algorithm (LGA) was used and dimensions of the grids were set at $60 \times 60 \times 60$ Å in the x, y and z directions, with a spacing of 0.375 Å between the grid points and the center placed at the active site of TLR4-MD-2 Complex crystallographic structures. All other docking and consequent scoring parameters used were kept at their default settings. The docked conformation corresponding to the lowest binding energy was selected as the most probable binding conformation.

**Statistical analysis**. The results for statistical analysis are presented as the mean ± standard error mean (SEM). The statistical differences between groups were tested by one-way analysis of variance (ANOVA) for multiple comparisons followed by Tukey's post hoc test using GraphPad Prism software (version 7.0, La Jolla, CA, USA). $P$-value < 0.05 was considered statistically significant (*$p < 0.05$, **$p < 0.01$, ***$p < 0.001$).

## Data availability
The authors declare that all data which supports the findings are provided with the paper and its supplementary information. All data is available from the corresponding authors upon reasonable request.

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

## Acknowledgements

This work was supported by the National Natural Science Foundation of China (Nos. 81303074, 81673719, 21473257, and 21773311), the Outstanding Youth Foundation of Hunan Provincial Natural Science Foundation of China (No. 2019JJ30042), the Hunan Provincial Natural Science Foundation of China (No. 2019JJ50960), and the Special Program for Applied Research on Super Computation of NSFC-Guangdong Joint Fund. We are very grateful to Prof. Zhimou Yang from NanKai University for giving us guidance on experimental methods of using LC-MS to evaluate the stability of rhein gel in cells.

## Author contributions

Y.W., Y.Z., R.F. and T.T. conceived and designed the research. J.Z. prepared rhein hydrogel and investigated the self-assembly mechanism of the rhein hydrogel. R.F., Y.Y., H.Y., J.L., P.Z. and P.G. conducted the study of cytotoxic and anti-inflammatory mechanisms of rhein hydrogel. H.W., T.Y. and L.D. carried out the DFT theoretical calculation. J.Z. performed the spectroscopic characterizations. L.Y. and L.R. performed mass experiments. Z.S. prepared and performed data collection. J.Z., Y.W., R.F. and Y.Y. and drafted and revised the manuscript. J.Z., Y.Z. and Y.W. wrote Supplementary information.

## Additional information

**Competing interests:** The authors declare no competing interests.

