## [Peer Review File · Nature Communications]

Reviewers' comments:

Reviewer #1 (Remarks to the Author):

I have read with great interest the manuscript by Zhang et al. The manuscript describes the fabrication of hydrogel based on the self assembly of a small molecule (named rhein). The hydrogels have been thoroughly characterized by a range of techniques to confirm the mechanism of the formation of the hydrogels as well the physical and mechanical properties. The potential use of the hydrogels has also been assessed and, due to the low toxicity, the materials can be applied in health. Overall, this is an interesting manuscript. However, the manuscript seems to fit a more specialized journal rather than a journal of broad audience such a Nature Communications.

Minor Comments

1) The manuscript has a lot of typos (including the title), so it is imperative to carefully check and correct those before further submission. Some of the sentence structures need to be improved for clarity.

2) The gels seems to be formed by self assembly via the hydrogen bonding and pi-pi interactions. If this is the case, the gel may have some self healing properties. Has the authors looked into this?

Reviewer #2 (Remarks to the Author):

This manuscript by Zhang and coworkers demonstrated the direct self-assembled hydrogel of a nature small molecule rhein. The rhein hydrogel increases the accumulation to bind to TLR4 strongly, thus alleviating neuroinflammatory responses through TLR4/NFκB pathway. Moreover, the rhein hydrogel exhibits better inhibition of TLR4/NFκB pathway than the free drug. This work represents a novel application of hydrogel and the relevant of supramolecular assemblies, thus I support the acceptance of this work after the authors revise their manuscript to address the following issues.

1. The reversibility of rhein hydrogel should be proved using rheometer.
2. The authors claimed the biostability and prevention of degradation of rhein hydrogel, but there is no result to support this claim, I suggest the authors to compare the biostability and metabolism of rhein hydrogel and free rhein in live cells.
3. Figure 3b, why are the data points for 4 mg/mL rhein gel missing after 36 h?
4. Scheme 1 appears incorrect. The TLR4/NFκB pathway shouldn't be blocked under inflammatory response (right panel). The authors should clarify this.
5. Some of the references are not cited properly, i.e., ref 11 is not discussing drug carriers and ref 15 is not related to taxol. For self-assembling taxol hydrogel, *J. Am. Chem. Soc.*, 2009, 131 (38), pp 13576–13577 and *Biomaterials* 33 (24), 5848-5853 should be cited.
6. The authors should proof-read the manuscript to correct the grammar errors, like "promissing" and "effiency" in the title and Line 94 "In particular, the as-prepared rhein hydrogel attenuates neuroinflammatory responses is better than that of its free drug from and almost no cytotoxicity."

Reviewer #3 (Remarks to the Author):

This is a very interesting manuscript on the self-assembly of natural small molecule into hydrogel. The concept of 'ideal drug gels' is nice and the use of rhein in a simple, biocompatible assembly process is indeed attractive. Moreover, the physical, chemical and biological properties of the macroscopic end-product are intriguing.

I have only a few issues that should be addressed:

1. Scheme 1 should be much better explained, both in the text as well as in the figure legends. Indeed the formation of tetramers and their stability is explained later but it is out of context in scheme 1.
2. With all the advantages of DFT, it is not a definite proof for the formation of dimers and tetramers. Ion Spray Mass Spectrometry or other techniques should give direct experimental data.
3. Furthermore with relationship to the dimer/tetramer/fiber organization: Is there crystal structure of rhen? It could be useful to understand the molecular details of the organization.
4. I agree that the assembly process of the natural molecules under ambient conditions is attractive. I feel about uncomfortable with the "green process" hype. It is exactly the same straightforward process as used to make gels based on agarose, gelatin, or other biopolymers.

Minor issues:

1. "Dhenylalanine" should be "Diphenylalanine" (line 68)
2. "Nano. Today" should be "Nano Today" (line 514)
3. " Nature Nanotech" should " Nat. Nanotech." (line 526)

Answers to referees' questions

Point-by-point response to the individual questions of Referee 1:

Reviewer #1 (Remarks to the Author):

Q overall: I have read with great interest the manuscript by Zhang et al. The manuscript describes the fabrication of hydrogel based on the self-assembly of a small molecule (named rhein). The hydrogels have been thoroughly characterized by a range of techniques to confirm the mechanism of the formation of the hydrogels as well the physical and mechanical properties. The potential use of the hydrogels has also been assessed and, due to the low toxicity, the materials can be applied in health. Overall, this is an interesting manuscript. However, the manuscript seems to fit a more specialized journal rather than a journal of broad audience such a Nature Communications.

A: Thank you very much for your important suggestions. After carefully reading our manuscript, we believe that this work is suitable to be published in Nature Communications. This study is full of multidisciplinary cross connection. We explored a Direct self-assembly hydrogel of natural small molecules from herbal medicine. Moreover, this novel hydrogel enhance protection against neuroinflammatory disorders. It is notable that this work combines chemical advancement with medicinal application. Amounts of similar integrative studies have been broadly published in Nature Communications:

1. Yu M, et al. Rapid transport of deformation-tuned nanoparticles across biological hydrogels and cellular barriers. **Nature Communications**. 2018, 9(1): 2607.

2. Mateen R, et al. A printable hydrogel microarray for drug screening avoids false positives associated with promiscuous aggregating inhibitors. **Nature Communications**. 2018, 9(1): 602.
3. Hong LTA, et al. An injectable hydrogel enhances tissue repair after spinal cord injury by promoting extracellular matrix remodeling. **Nature Communications**. 2017, 8(1): 533.
4. Liu J, et al. Triggerable tough hydrogels for gastric resident dosage forms. **Nature Communications**. 2017, 8(1): 124.
5. Rape AD, et al. A synthetic hydrogel for the high-throughput study of cell-ECM interactions. **Nature Communications**. 2015, 6: 8129.

According to your constructive suggestions, we have carefully rewritten the introduction section. The novelties of this contribution are highlighted: This is the novel injectable gel which is formed through a simple and facile treatment with a 100% pure natural product. And also, the significances of this contribution are re-emphasized: 1. Our experimental results confirmed that the fibrils of rhein exhibit lower toxicity, higher medical activity and better stability than its monomer against neuroinflammation in vitro. 2. Rhein itself is not soluble, therefore it was usually embedded into a carrier or cargo to enter the stomach of human. However, drug loss during the fabrication process and premature release of payload still lead to lower drug loading and adverse systemic toxicity. Hence, the injection of rhein gel seems to be the most promising administration route of such kind of natural small molecular drugs. 3. This strategy can be extended to similar natural products.

After revising the paper according to your kind suggestions, we now believe that this paper will benefit a broader range of readers of Nature Communications.

Minor Comments

Q1: 1) The manuscript has a lot of typos (including the title), so it is imperative to carefully check and correct those before further submission. Some of the sentence structures need to be improved for clarity.

A: According to your suggestion, we had spent a lot of time in working with the English professionals on campus to correct grammar, syntax, and flow; and subsequently sent the draft to several native speakers to further polish English before re-submitting. Furthermore, we also search help with English Language Editing from Springer Nature. The revised details are marked in red color in the revision. The edit certificate is as follows:

SPRINGER NATURE | Author Services

Nature Research Editing Service Certification

This is to certify that the manuscript titled Direct self-assembly hydrogel of herbal small molecule: a promising way toward high efficient natural medicines was edited for English language usage, grammar, spelling and punctuation by one or more native English-speaking editors at Nature Research Editing Service. The editors focused on correcting improper language and rephrasing awkward sentences, using their scientific training to point out passages that were confusing or vague. Every effort has been made to ensure that neither the research content nor the authors' intentions were altered in any way during the editing process.

Documents receiving this certification should be English-ready for publication; however, please note that the author has the ability to accept or reject our suggestions and changes. To verify the final edited version, please visit our verification page. If you have any questions or concerns over this edited document, please contact Nature Research Editing Service at support@as.springernature.com.

Manuscript title: Direct self-assembly hydrogel of herbal small molecule: a promising way toward high efficient natural medicines

Authors: Jun Zheng, Rong Fan, Huiqiong Wu, Yujie Yan, Lu Ran, Zhifang Sun, Lunzhao Yi, Li Dang, Pingping Gan, Honghui Yao, Piao Zheng, Tilong Yang, Yang Wang, Tao Tang, Yi Zhang

Key: 89B0-E870-B090-0535-2F38

This certificate may be verified at secure.authorservices.springernature.com/certificate/verify.

Q2: 2) The gels seems to be formed by self-assembly via the hydrogen bonding and pi-pi interactions. If this is the case, the gel may have some self-healing properties. Has the authors looked into this?

A: Thank you very much for your intensive comprehension over our manuscript. The Referee is right, the gel does have self-healing properties. We have performed oscillatory shear rheology to confirm the self-healing properties:

According to your suggestions, the self-healing properties of the rhein hydrogel were investigated by oscillatory shear rheology (Fig. 1). As shown in figure 1e, when the strain exceeds 23.74%, The loss modulus (G'') was higher than (G'), indicating the transition of the gel state to solution state (solution state: $G' < G''$, gel state: $G' > G''$). The step-strain test revealed that G' was higher than G'' at low strain 0.1%, G' was lower than G'' at higher strain 35%, then the material properties of the rhein hydrogel recovered rapidly when transitioning from high strain to low strain (Fig.1f), indicating the injectable and self-healing properties of the rhein hydrogel.

Fig. 1 Self-healing properties of rhein hydrogel. (e) Strain-dependent oscillatory shear rheology of the rhein hydrogel with a fixed frequency of $10 \text{ rad} \cdot \text{s}^{-1}$, $T=25^\circ\text{C}$. (f) Step-strain measurements of the hydrogel over five cycles at low strain (0.1%) and

high strain (35%), $T=25^{\circ}\text{C}$, frequency $10\ 10\ \text{rad}\cdot\text{s}^{-1}$.

The above results are added to the manuscript (shown in Fig. 1e & 1f). Many thanks for your kind suggestions.

Point-by-point response to the individual questions of Referee 2:

Q overall: This manuscript by Zhang and coworkers demonstrated the direct self-assembled hydrogel of a nature small molecule rhein. The rhein hydrogel increases the accumulation to bind to TLR4 strongly, thus alleviating neuroinflammatory responses through TLR4/NF κ B pathway. Moreover, the rhein hydrogel exhibits better inhibition of TLR4/NF κ B pathway than the free drug. This work represents a novel application of hydrogel and the relevant of supramolecular assemblies, thus I support the acceptance of this work after the authors revise their manuscript to address the following issues.

A: Thank you so much for your positive and constructive suggestions. We have revised the manuscript according to your next suggestions.

Q1: The reversibility of rhein hydrogel should be proved using rheometer.

A: Thank you very much for giving us such a good suggestion. We have performed oscillatory shear rheology to confirm the reversibility of rhein hydrogel.

According to your suggestions, the reversible properties of the rhein hydrogel were investigated by oscillatory shear rheology (Fig. 1). As shown in figure 1a, when the strain exceeds 23.74%, The loss modulus (G'') was higher than (G'), indicating the

transition of the gel state to solution state (solution state: $G' < G''$, gel state: $G' > G''$). The step-strain test revealed that G' was higher than G'' at low strain 0.1%, G' was lower than G'' at higher strain 35%, then the material properties of the rhein hydrogel recovered rapidly when transitioning from high strain to low strain (Fig.1f), indicating the injectable and self-healing properties of the rhein hydrogel.¹⁻³

Fig. 1 Rheologic characterization of rhein hydrogel. (e) Strain-dependent oscillatory shear rheology of the rhein hydrogel with a fixed frequency of $10 \text{ rad}\cdot\text{s}^{-1}$, $T=25^\circ\text{C}$. (f) Step-strain measurements of the hydrogel over five cycles at low strain (0.1%) and high strain (35%), $T=25^\circ\text{C}$, frequency $10 \text{ rad}\cdot\text{s}^{-1}$.

The above results are added to the manuscript (shown in Fig. 1e & 1f). Thank you again for reminding us to perform this test.

Q2 The authors claimed the biostability and prevention of degradation of rhein hydrogel, but there is no result to support this claim, I suggest the authors to compare the biostability and metabolism of rhein hydrogel and free rhein in live cells.

A: Thank you for pointing out the weak points of this part. This a very constructive and of great importance for the suggestion.

According to your suggestion, after a thoughtful discussion with Prof. Zhimou Yang@State Key Laboratory of Medicinal Chemical Biology, Nankai University, he gave us a very inspirational suggestion of using liquid chromatography-mass spectrometry (LC-MS) to identify the biostability and metabolism of rhein hydrogel. According to this constructive suggestion, we used LC-MS method to detect the biostability and metabolism of rhein hydrogel in BV2 cells. The cells were cultured with rhein hydrogel and free-drug for 3 h, 12 h, 24 h, 48 h, and 72 h, respectively.

Fig. 3 Time elapsing concentration changes of total rhein in the cells was cultured with rhein hydrogel and free-drug, respectively.

Supplementary Figure 12. Representative Multiple reaction monitoring (MRM) chromatogram of rhein and dimer in BV2 cells cultured with rhein hydrogel and free-drug. (a) rhein standard solution (b) concentration ratio of dimer to rhein monomer in cells cultured with hydrogel. (c) Rhein group 3 h. (d) Rhein hydrogel group 3 h. (e) Rhein group 12 h, (f) Rhein hydrogel group 12 h, (g) Rhein group 24 h. (h) Rhein hydrogel group 24 h. (i) Rhein group 48 h. (g) Rhein hydrogel group 48 h.

As shown in Fig. 3b, the concentration of the rhein from hydrogel group was higher than the free-drug group at each time point. And it is notable that we detected the presence of dimers in the cells incubated with rhein hydrogel, while in the sample of free drug, no dimers were detected (Supplementary Fig. 12). These data indicate that rhein nanofiber are more likely to penetrate into cells rather than formed from free-drug in the cells. In addition, we observed that the concentration of free-drug reached the top at 24 h and then decreased at 48 h. While there was a plateau during 24 to 48 h in rhein hydrogel group, indicating that the rhein hydrogel is more stable in the cells.

Q3. Figure 3b, why are the data points for 4 mg/mL rhein gel missing after 36 h?

A: Thank you for pointing out this confusion issue. We have re-examined the sustained release of rhein hydrogel at a series of concentrations. Fig. 3a confirmed the release profile. There was a fast release during the first 12 h, and then the release of rhein showed a gradual process. Regardless of high concentration or low

concentration, the release rate was up to 70% after 24 h, which was attributed the good solubility of the hydrogel and the brittleness of the fibers. As the gel concentration increased, the release rate decreased, accompanied by the extension of the release time. Moreover, the cumulative release percentage at low concentrations reached 90% within 36 h, and merely a small amount of rhein can be released from 36 h to 72 h. The cumulative release rate with high concentrations merely came to 80% within 36 h and the release rate reached 88% after sustained released to 72 h.

Fig. 3 In vitro release profiles of rhein hydrogel. The concentration of rhein hydrogel was $4 \text{ mg}\cdot\text{mL}^{-1}$ (14 mM) and $6 \text{ mg}\cdot\text{mL}^{-1}$ (21 mM), respectively.

Q4. Scheme 1 appears incorrect. The TLR4/NFκB pathway shouldn't be blocked under inflammatory response (right panel). The authors should clarify this.

A: Thank you for pointing out this ambiguity. We have rearranged the Scheme to correct the errors, please see the revised scheme for the details:

Scheme 1. Schematic depiction of the self-assembly process of rhein hydrogel and neuroinflammatory prevention by the rhein hydrogel in BV2 microglia cells. Rhein is self-assembled by non-covalent interaction forces to form nanofibers with left-handed, which are further crosslinked to form 3D network structure. When treating inflammatory BV2 cells with rhein nanofibers. The nanofiber slowly depolymerize to release rhein or rhein aggregates. They intensively bind to MD-2 (the subunit of TLR4 receptor), obscure the active site of TLR4 and further block the access of the substrate, leading to the inhibition of NFκB activation. Subsequently, the rhein hydrogel suppresses the release of neuroinflammatory factors and mediators.

Q5. Some of the references are not cited properly, i.e., ref 11 is not discussing drug carriers and ref 15 is not related to taxol. For self-assembling taxol hydrogel, J. Am. Chem. Soc., 2009, 131 (38), pp 13576–13577 and Biomaterials

33 (24), 5848-5853 should be cited.

A: Thank you very much for pointing out the mistakes of citations. We are sorry for the careless errors. And thank you again for introducing these representative literatures in this field. Your extensive knowledgeability is so helpful for us to improve the quality of this paper. Meanwhile, we've appended all these mentioned literatures to our reference library.

16 Yuan, G. *et al.* Enzyme-instructed molecular self-assembly confers nanofibers and a supramolecular hydrogel of taxol derivative. *J. Am. Chem. Soc.* **131**, 13576-13577 (2009).

17 Wang, H. *et al.* The inhibition of tumor growth and metastasis by self-assembled nanofibers of taxol. *Biomaterials* **33**, 5848-5853 (2012).

Q6 The authors should proof-read the manuscript to correct the grammar errors, like “promissing” and “effiency” in the title and Line 94 “In particular, the as-prepared rhein hydrogel attenuates neuroinflammatory responses is better than that of its free drug from and almost no cytotoxicity.”

A: Thank you for your kind suggestions, according to your suggestion we had worked with the English professionals on campus to correct grammar, syntax, and flow; and subsequently sent the draft to several native speakers to further polish English before re-submitting. We also search help with English Language Editing from Springer Nature. The edit certificate is as follows:

Nature Research Editing Service Certification

This is to certify that the manuscript titled *Direct self-assembly hydrogel of herbal small molecule: a promising way toward high efficient natural medicines* was edited for English language usage, grammar, spelling and punctuation by one or more native English-speaking editors at Nature Research Editing Service. The editors focused on correcting improper language and rephrasing awkward sentences, using their scientific training to point out passages that were confusing or vague. Every effort has been made to ensure that neither the research content nor the authors' intentions were altered in any way during the editing process.

Documents receiving this certification should be English-ready for publication; however, please note that the author has the ability to accept or reject our suggestions and changes. To verify the final edited version, please visit our verification page. If you have any questions or concerns over this edited document, please contact Nature Research Editing Service at support@as.springernature.com.

Manuscript title: Direct self-assembly hydrogel of herbal small molecule: a promising way toward high efficient natural medicines

Authors: Jun Zheng, Rong Fan, Huiqiong Wu, Yujie Yan, Lu Ran, Zhifang Sun, Lunzhao Yi, Li Dang, Pingping Gan, Honghui Yao, Piao Zheng, Tulong Yang, Yang Wang, Tao Tang, Yi Zhang

Key: 89B0-E870-B090-0535-2F38

This certificate may be verified at secure.authorservices.springernature.com/certificate/verify.

Point-by-point response to the individual questions of Referee 3

Q overall: This is a very interesting manuscript on the self-assembly of natural small molecule into hydrogel. The concept of ‘ideal drug gels’ is nice and the use of rhein in a simple, biocompatible assembly process is indeed attractive. Moreover, the physical, chemical and biological properties of the macroscopic end-product are intriguing.

A: Thank you very much for your kind suggestions. We hope that the revision could be published in Nature Communications.

Q1. Scheme 1 should be much better explained, both in the text as well as in the figure legends. Indeed the formation of tetramers and their stability is explained later but it is out of context in scheme 1.

A: Thank you so much for this very constructive suggestion. We totally agree with you. A clearly introductory figure will help a lot to provide the reader much clearer understand about the key findings of this paper. According to your suggestion, Scheme 1 has been re-plotted to provide an introductory explanation of the most important discoveries of this paper.

Scheme 1. Schematic depiction of the self-assembly process of rhein hydrogel and neuroinflammatory prevention by the rhein hydrogel in BV2 microglia cells. Rhein is self-assembled by non-covalent interaction forces to form nanofibers with

left-handed, which are further crosslinked to form 3D network structure. When treating inflammatory BV2 cells with rhien nanofibers. The nanofiber slowly depolymerizes to release rhein or rhein aggregates. They intensively bind to MD-2 (the subunit of TLR4 receptor), obscure the active site of TLR4 and further block the access of the substrate, leading to the inhibition of NF κ B activation. Subsequently, the rhein hydrogel suppresses the release of neuroinflammatory factors and mediators.

Q2. With all the advantages of DFT, it is not a definite proof for the formation of dimers and tetramers. Ion Spray Mass Spectrometry or other techniques should give direct experimental data.

A: We appreciate this GREAT suggestion. It needs to point out that according to your suggestion, we have used Ion Spray Mass Spectrometry to reexam the possibility oligomers in diluted. The presences of tetramers, pentamers, hexamers and heptamers have been demonstrated. And the particular meaning of this finding is also discussed in the text. Here we offer our sincere appreciate in advance for your further valuable suggestions in this scheme. We have observed the presence of tetramers, pentamers, hexamers and heptamers in the rhein gel dilution, please see the figures below.

Fig. 2 Mass spectrometry (MS) analyses of the rhein nanofiber.

Supplementary Figure 8. MS/MS analysis of the rhein nanofiber. (a) MS/MS

analyses of monomer [M-H]⁻, **(b)** MS/MS analyses of dimer [2M-2H+Na]⁻, **(c)**

MS/MS analyses of trimer $[3M-3H+2Na]^-$, (d) MS/MS analyses of tetramer $[4M-4H+3Na]^-$, (e) MS/MS analyses of pentamer $[5M-5H+4Na]^-$.

Our experimental results confirm the presence of the monomers (m/z 283.0249), dimers (m/z 589.0388), trimers (m/z 895.0514), tetramers (m/z 1201.0647). Higher-order aggregates were observed in mass spectrum (Fig. 3a and 3b), and these aggregates existed as sodium salt clusters. The dimers were the predominant multimer ions, indicating that the dimer is relatively stable relative to other aggregates. These results were further proved by the MS/MS analysis, we found the fragment ions of the dimer, rhein sodium and deprotonated rhein in the MS/MS spectrum of higher-order aggregates. These results indicate that rhein monomers and sodium salt of rhein are the building block for rhein nanofibers, rhein monomer and sodium rhein together assemble to form dimer and higher-order aggregates. We previously thought that the dimer configuration was formed by the carboxylic group of each monomer interacts with the other via intermolecular hydrogen bonds, which is inconsistent with the information obtained by mass spectrometry. Therefore, based on the experimental results, we re-proposed the possible configurations of the aggregates. The two molecules are arranged in a J-type aggregation manner by π - π stacking and hydrogen bonding to form a dimer. Due to the electrostatic repulsion between the carboxylic acid ions, two molecules are arranged in opposite direction. Subsequently, dimers were further assembled into trimers, tetramers and higher-order aggregates.

We strongly believe the presence of even larger oligomers, and we are very regret

to be informed that the detection range of this Ion Spray/ambient ionization mass spectrometry is limited. The molecular weight of heptamer/ 6Na^+ is 2119.1057, the largest oligomer we can observe so far.

Q3. Furthermore, with relationship to the dimer/tetramer/fiber organization: Is there crystal structure of Rhein? It could be useful to understand the molecular details of the organization.

A: Thank you for your constructive suggestion. You are right, the crystal structure of this small molecule is of key importance to understand the self-assembly process of Rhein hydrogel. Unfortunately, we have been not able to obtain the perfect single crystal of Rhein to date. According to your suggestion, we did our best to use X-ray diffraction (XRD) and Mass Spectrometry (MS) to understand the structural information of Rhein aggregates.

Supplementary Figure 8. X-ray diffraction (XRD) pattern of the Rhein xerogel (the unit of inserted distances is Å).

The XRD pattern of the rein xerogel has five diffraction peaks at $2\theta = 22.4^\circ$, 23.3° , 26.1° , 31.4° , 33.1° , corresponding to distance of 4.0, 3.8, 3.4, 2.8 and 2.7 Å, respectively (Supplementary Fig. 8). The distance of 4.0 Å is assigned to the stacking distance of the rein molecular backbones.⁴ The apparent peak at $d = 3.4$ Å is a typical distance of π - π stacking interactions between the two molecular.⁵ The reflection at 2.8 Å is the distance between neighboring rein molecular.⁶ The XRD data of rein xerogel showed an imperfect polycrystalline structures, providing very poor information about the crystal structure of rein. Therefore, based on the experimental results, we speculated on the self-assembly process of rein nanofibers. As shown in picture 2e, the two molecules are arranged in a J-type aggregation manner by π - π stacking and hydrogen bonding to form a dimer. Due to the electrostatic repulsion between the carboxylic acid ions, two molecules are arranged in opposite direction. Subsequently, dimers were further assembled into trimers, tetramers and higher-order aggregates. The rein molecules continually added to the per-existing aggregates in a left helix fashion, resulting in the formation of the nanofiber with a left-handed helical configuration. The nanofibers further crosslink to form 3D networks (Fig. 2e).

Fig. 2 Self-assembly process diagram of the rein hydrogel.

Q4. I agree that the assembly process of the natural molecules under ambient conditions is attractive. I feel about uncomfortable with the "green process" hype. It is exactly the same straightforward process as used to make gels based on agarose, gelatin, or other biopolymers.

A: Thank you for your consideration. Yes, you are right. After discussion with the authors, we have decided to delete the information on green process in main text and title. The manuscript title was revised as “Direct self-assembly hydrogel of herbal small molecule: a promising way toward high efficient natural medicines”.

Minor issues:

1. "Dhenylalanine" should be "Diphenylalanine" (line 68)

A: Yes, Thank you. We have revised it in manuscript.

2. "Nano. Today" should be "Nano Today" (line 514)

A: Yes, Thank you. We have revised it in manuscript.

3. " Nature Nanotech" should " Nat. Nanotech." (line 526)

A: Yes, Thank you. We have revised it in manuscript.

References:

1 Lyu, D., Chen, S. & Guo, W. Liposome crosslinked polyacrylamide/DNA

hydrogel: a smart controlled-release system for small molecular payloads.

Small **14**, 1704039-1704046 (2018).

2 Mealy, J. E. *et al.* Injectable granular hydrogels with multifunctional properties for biomedical applications. *Adv. Mater.* **30**, 1705912-1705917 (2018).

3 Liu, Q. *et al.* A supramolecular shear-thinning anti-inflammatory steroid hydrogel. *Adv Mater* **28**, 6680-6686 (2016).

4 Wang, X. *et al.* Designing isometrical gel precursors to identify the gelation pathway for nickel-selective metallohydrogels. *Dalton Trans.* **45**, 18438-18442 (2016).

5 Liu, K. *et al.* Coordination-triggered hierarchical folate/zinc supramolecular hydrogels leading to printable biomaterials. *Acs Appl. Mater. Interfaces* **10**, 4530-4539 (2018).

6 Castelletto, V. *et al.* Influence of elastase on alanine-rich peptide hydrogels. *Biomater. Sci.* **2**, 867-874 (2014).

REVIEWERS' COMMENTS:

Reviewer #1 (Remarks to the Author):

I am read the responses and the revised version of the manuscript, it is now suitable for publication in Nature Communications.

Reviewer #2 (Remarks to the Author):

The authors have addressed my previous concerns. I support the acceptance of this work.

Reviewer #3 (Remarks to the Author):

The revised manuscript is much improved. The authors had addressed all my concerns as well as those of the other reviewers.